# Spontaneous Tumors and Non-Neoplastic Proliferative Lesions in Pet Degus (*Octodon degus*)

**DOI:** 10.3390/vetsci7010032

**Published:** 2020-03-13

**Authors:** Tanja Švara, Mitja Gombač, Alessandro Poli, Jožko Račnik, Marko Zadravec

**Affiliations:** 1Institute of Pathology, Wild Animals, Fish and Bees, Veterinary Faculty, University of Ljubljana, SI-1000 Ljubljana, Slovenia; Tanja.Svara@vf.uni-lj.si (T.Š.); Mitja.Gombac@vf.uni-lj.si (M.G.); 2Department of Veterinary Science, University of Pisa, 56124 Pisa, Italy; 3Clinic for Birds, Small mammals and Reptiles, Institute of Poultry, Birds, Small Mammals and Reptiles, Veterinary Faculty, University of Ljubljana, SI-1000 Ljubljana, Slovenia; Josko.Racnik@vf.uni-lj.si (J.R.); marko_zadravec@yahoo.com (M.Z.)

**Keywords:** degu, non-neoplastic proliferative lesions, *Octodon degus*, pathology, tumors

## Abstract

In recent years, degus (*Octodon degus*), rodents native to South America, have been becoming increasingly popular as pet animals. Data about neoplastic diseases in this species are still sparse and mainly limited to single-case reports. The aim of this study was to present neoplastic and non-neoplastic proliferative changes in 16/100 pet degus examined at the Veterinary Faculty University of Ljubljana from 2010 to 2015 and to describe the clinic-pathological features of these lesions. Twenty different lesions of the integumentary, musculoskeletal, genitourinary and gastrointestinal systems were diagnosed: amongst these were 13 malignant tumors, six benign tumors, and one non-neoplastic lesion. Cutaneous fibrosarcoma was the most common tumor (7/16 degus). It was detected more often in females (6/7 degus) and lesions were located mainly in hind limbs. The gastrointestinal tract was frequently affected, namely with two malignant neoplasms - an intestinal lymphoma and a mesenteric mesothelioma, four benign tumors – two biliary cystadenomas, an oral squamous papilloma and a hepatocellular adenoma, and a single non-neoplastic proliferative lesion. In one animal, two organic systems were involved in neoplastic lesions.

## 1. Introduction

The common degu (*Octodon degus*) is a rodent of the Octodontidae family endemic to Central Chile, where it inhabits the open savannah [1]. Degus live well in the laboratory environment and are commonly used in animal models to study behavior and brain function, particularly social behavior, circadian rhythms, reciprocal kinship and endocrinology [2]. Recently, this species has been used to investigate other questions regarding reproduction, diabetes mellitus, and particularly, Alzheimer’s disease because it can spontaneously develop cognitive decline with concomitant phospho-tau, β-amyloid pathology [3,4]. In recent years, this species has been becoming popular as pet animals [5]. Descriptions of degus diseases are rare and the incidence of neoplasia appears to be low [5]. Other than single case reports [6,7,8,9,10,11,12,13], seven neoplasms were detected in 189 animals [14] and six tumours were described in another study performed in 300 degus [5]. To date, no reports about non-neoplastic proliferative lesions in degus have been reported. The aim of this study was to describe the clinicopathological findings of spontaneous tumors and other proliferative lesions detected in 16 pet degus examined at the Veterinary Faculty University of Ljubljana from 2010 to 2015.

## 2. Materials and Methods

### 2.1. Subjects

Between 2010 and 2015, 100 pet degus were referred at the Institute of Poultry, Birds, Small Mammals and Reptiles, Veterinary Faculty, University of Ljubljana. Among these, sixteen subjects (ten females and five males; gender information was missing for one animal) presented neoplastic or non-neoplastic proliferative lesions. The age of these degus ranged from 3 to 9 years (mean 6 ± 1.9 years). 

### 2.2. Pathological Investigations

A complete necropsy was performed on twelve subjects and biopsy samples taken from the other four degus. Tissue samples were fixed in 10% neutral buffered formalin and routinely processed for paraffin embedding. Four-micrometers thick sections were stained with Hematoxylin and Eosin, Goldner’s Trichrome and Periodic-Acid Schiff. Immunohistochemistry was performed on the selected sections using mouse monoclonal antibodies anti-human vimentin (1:50; clone V9, Dako, Glostrup, Danmark), anti-human cytokeratin AE1/AE3 (1:100; Dako, Glostrup, Danmark), desmin (1:50; clone D33, Dako, Glostrup, Danmark) and anti-human c-kit (CD117) (1:300; Dako, Glostrup, Danmark). Antibody binding was detected by EnVision Detection System Peroxidase/DAB +, Rabbit/Mouse (DAKO, Glostrup, Danmark). Tumours were classified according to the WHO classifications of tumors of domestic animals [15,16,17,18]. 

## 3. Results

Twenty different lesions were detected. Neoplastic and other non-neoplastic proliferative lesions were detected in the integumentary, musculoskeletal, genitourinary and gastrointestinal systems (Table 1). In one degu (6.2% of examined subjects), more than one organic system was involved.

Cutaneous and subcutaneous tumors were detected in 7/16 (if considering individual degus) or 8/20 (when considering individual lesions). Cutaneous fibrosarcoma was detected in seven degus (six females and one male) (degus # 1–7). The age of animals with fibrosarcoma ranged from 3 to 9 years with the mean age of 6 ± 1.9 years. Grossly, these tumors were firm masses measuring from 1 to 3 cm in diameter and with a grey-white cut surface. The most common localization was hind limbs (N = 4/7) (Figure 1A), but these were also detected in the auricular region, the back region and on the tail. At necropsy, no metastases were detected in regional lymph nodes or other organs. Histopathological examination revealed poorly circumscribed, infiltrative, non-encapsulated neoplastic mass composed of bundles of spindle cells. Neoplastic cells showed moderate-to-severe anisocytosis and anisokaryosis, variable mitotic index (from 5 to 50 mitoses per 10 high power fields) and variable extension of necrotic areas. Single bi- and multinucleated cells were also observed (Figure 2A). No blood or lymph vessels invasion was observed. According to the criteria for grading soft tissue sarcomas in dogs [19], four tumors were graded as moderately differentiated (intermediate grade), two were graded as poorly differentiated (high-grade) and one as well differentiated (low-grade). Some degus with fibrosarcoma were also closely related. Indeed, fibrosarcomas were detected in two mothers (degus # 1 and 6), and their female offspring (degus # 2, 5 and 7). Subcutaneous lipoma was detected in an 8-year-old female subject (degu # 5) and was presented as an oval, well-circumscribed, non-encapsulated soft tumor on the left flank measuring 2 cm in diameter. The tumor was composed of well differentiated neoplastic cells resembling normal adipose tissue. 

Bone tumors were observed in two 5-years-old degus (degus # 8 and # 9), while non-neoplastic lesions were not observed. Chordoma was diagnosed in a male (degu # 8) presented with a firm mass on the tail measuring 1 cm (Figure 1B). Histopathologically, well-circumscribed, non-encapsulated tumor consisted of physaliferous cells with vacuolated cytoplasm and round, eccentric nuclei exhibiting anisocytosis. No mitoses or blood and lymph vessel invasion were observed (Figure 2B). Degu had no recurrence 16 months after surgery. Femoral osteosarcoma was detected in a female (degu # 9) with a large firm mass on the left hind limb. The degu underwent amputation of the affected hind limb but died shortly after surgery. Numerous round white multifocal lesions up to 2 mm in diameter were scattered also in the pulmonary parenchyma. The histopathology of the limb mass revealed a poorly circumscribed, infiltrative, non-encapsulated densely cellular neoplasia arising from the femur. Bundles of spindle, oval and round cells of various sizes, with oval and round, anisokaryotic nuclei with large eosinophilic nucleoli were detected. Single multinucleated cells, bizarre nuclei and multifocal areas of coagulative necrosis (less than 50% of the tumor) were also observed. No osteoid production was observed and the tumor was diagnosed as non-productive osteoblastic osteosarcoma. Pulmonary lesions were consistent with multifocal metastases consisting of sarcomatous pleomorphic neoplastic cells exhibiting anisocytosis and anisokaryosis. 

Genitourinary neoplastic lesions were detected in two female degus. Degu # 10 had two asynchronous genitourinary tumors, and degu # 11 also had also renal haemangioma concomitant with a mesenteric mesothelioma and a hepatic nodular hyperplasia. 

A uterine leiomyosarcoma measuring 2 cm in diameter was detected in a 4-year-old female (degu # 10). A well circumscribed, grey-white, firm, non-encapsulated neoplastic mass composed of spindle cells resembling normal smooth muscle cells was observed. Neoplastic cells showed moderate nuclear pleomorphism and anisokaryosis with bi- and multi-nucleated cells and multifocal areas of dystrophic calcification. The mitotic index was low. The animal died 2 years later and in the urinary bladder wall a round occluding, neoplastic mass measuring 1.5 cm in diameter was observed at necropsy. The histological examination revealed an infiltrative and poorly demarcated spindle cell tumor with marked nuclear pleomorphism, anisokaryosis and multifocal areas of dystrophic calcification and a diagnosis of a urinary bladder leiomyosarcoma was performed (Figure 2C). 

A renal cavernous hemangioma was diagnosed in an 8-year-old female (degu # 11) concomitant with mesenteric mesothelioma. Two dark red well circumscribed nodules located in the renal cortex and measuring 2 mm and 3 mm in diameter were composed of large vascular spaces lined by one layer of thin endothelial cells. 

Gastrointestinal lesions were frequently detected in the examined degus. Neoplastic lesions were observed in six animals, three males and two females (data about gender of one animal was not known). Non-neoplastic proliferative lesion was observed in one female animal (degu # 11) only. 

Mesenteric mesothelioma and hepatic nodular hyperplasia were diagnosed in degu # 11, euthanized due to persistent weight loss. At the necropsy, a large grey-white, firm neoplastic mass, infiltrating and accreting intestinal loops was found. Histologically, an infiltrative, non-encapsulated, dense cellular mass composed of spindle cells resembling fibroblast was detected. Neoplastic cells, which exhibited anisocytotosis and anisokaryosis, showed intense positive immunohistochemical reaction for CK AE1/AE3 (Figure 2D) and vimentin (Figure 2E) but were negative for desmin and c-kit (CD117) [20]. Moreover, a round, well-circumscribed hepatic mass, measuring 2 cm in diameter, was observed grossly. The histological examination revealed that a round, well-circumscribed, non-encapsulated lesion was formed of normal, but slightly disarranged hepatocytes. An oral squamous papilloma was diagnosed in a 3-year old female (degu # 12). A papillary grey-white firm exophytic mass measuring 3 mm in diameter was located on the palate behind maxillary incisors. Histologically, the exophytic tumor was covered with stratified squamous epithelium; no significant cytological atypia and no features suggestive of viral etiology have been detected. Intestinal lymphoma was diagnosed in a 6-year-old male (degu # 13) presented with signs of chronic weight loss. Segmental thickening of the small intestinal wall measuring up to 4 mm was detected. Histopathology revealed dense transmural infiltration of the small intestine wall and pancreas with lymphoid cells. Neoplastic cells were of medium size lymphocytes and had a low mitotic index (Figure 2F). 

Hepatocellular adenoma was diagnosed in a 7-year old male (degu # 14) with inappetence and reluctance to move. At necropsy, round, brown-red, firm hepatic mass, measuring 3 × 2 × 2 cm in diameter was detected. A well circumscribed, non-encapsulated lesion consisting of cords of well-differentiated neoplastic cells resembling hepatocytes, without portal triads and interlobular biliary ducts, was observed histologically. Biliary cystadenoma was diagnosed in two 8-year-old male degus (degus # 15 and 16). Solitary, well-circumscribed cystic lesions, measuring 3 mm and 7 mm were detected as incidental findings at necropsy. Microscopically, multifocal cystic structures lined by well differentiated cuboidal or columnar biliary epithelium were detected. 

## 4. Discussion

The tumor prevalence in pet degus is reported to be low [5,14]. Only seven cases of neoplasia were diagnosed in 189 laboratory degus (3.7%) [14], and a more recent study recorded six cases in 300 pet degus (2%) [5]. In our study, the prevalence of neoplastic and non-neoplastic proliferative disorders was higher (16%). Perhaps these data could be related to the older age of the examined animals (mean age 6 ± 1.9 years) vs. 2.3 ± 1.86 years in previous studies on pet degus [5], or because some of the animals of our study were referred to the university clinic for surgery. To date, different neoplastic lesions have been described in this species (cutaneous lipoma, melanoma, myxosarcoma, malignant histiocytoma, fibrosarcoma, cervical lymphosarcoma, hepatoma, hepatocellular carcinoma, splenic hemangioma, bronchioalveolar carcinoma, renal transitional cell carcinoma, renal choristoma, elodontoma, uterine angioleiomyoma, vaginal leiomyosarcoma, parathyroid adenocarcinoma, pulmonary adenocarcinoma and coccygeal chordoma) [6,7,8,9,10,11,12,13,14].

In our examined sample, tumors and proliferative lesions were found in several organic systems, but integumentary lesions prevailed, representing more than one third of diagnosed lesions. With the exception of one chordoma, mesenchymal tumors were detected our study, leading to the conclusion that degus are more prone to the development of cutaneous and subcutaneous mesenchymal tumors. To the author’s knowledge, epithelial tumors have not been reported in the integumentary system of degus. Cutaneous fibrosarcoma was the most common neoplastic integumentary lesion and the most common tumor type diagnosed in the present study. To date, only one case of fibrosarcoma has been reported in degu [5]. In our study, more cases were diagnosed in females than in males and hind limbs could be a preferential site. The age of animals with fibrosarcoma ranged from 3 to 9 years with a mean age of 6 ± 1.9 years. The average life span of degus is five to nine years [21], so fibrosarcomas appeared more frequent in older animals and it seems to affect genetically related animals. Although our preliminary results cannot allow definitive conclusions, recent investigations have shown that the pathogenesis of sarcomas is multifactorial including environmental and genetic components [22,23] and the possibility that sarcoma predisposition might be linked to genetic predisposition needs further investigation also in this species. Histologically, tumors showed various degree of differentiation. Using the criteria for grading canine soft tissue sarcomas [19], four fibrosarcomas were graded as moderately differentiated, two as poorly differentiated and one as well differentiated. In dogs, fibrosarcomas are generally of low-to-moderate malignancy and locally invasive. In degus, despite the prevalence of moderate and poorly differentiated fibrosarcomas with high mitotic rates (up to 50 mitoses per 10 high power fields), no metastases were detected in regional lymph nodes or other organs and tissues, suggesting that fibrosarcomas rarely metastasize in this species. The gastrointestinal tract was also frequently affected by neoplastic lesions in degus (n = 6; 37.5% of animals). Lesions were mainly detected in the liver but also in oral cavity, small intestine and mesentery. Differently from the integumentary system, where malignancy prevailed, almost all hepatic lesions were benign tumors. 

In our study, other tumors and non-neoplastic lesions were found as a single case. The morphology of these lesions resembles that described in other domestic animals [24]. Chordoma in degu resembled the features of chordoma in ferrets [25], differently from descriptions of rat chordoma located predominantly at lumbosacral vertebrae, with malignant features and associated with pulmonary metastases [26]. Interestingly, only two cases of genitourinary lesions were observed, both in females. In domesticated rats, mice and guinea pigs uterine and/or mammary gland disorders are quite common especially in aged animals [27]. Similarly to our findings in degus, the only other frequent domesticated rodent species that has low incidence of genitourinary lesions is chinchilla [28].

## 5. Conclusions

In conclusion, our study demonstrates that neoplastic lesions in degus, especially in older animals, can be very frequent. The most common are lesions of the integumentary system, and cutaneous fibrosarcoma is the most common neoplastic integumentary lesion and the most common tumor type diagnosed in the present study. 

## Figures and Tables

**Figure 1 vetsci-07-00032-f001:**
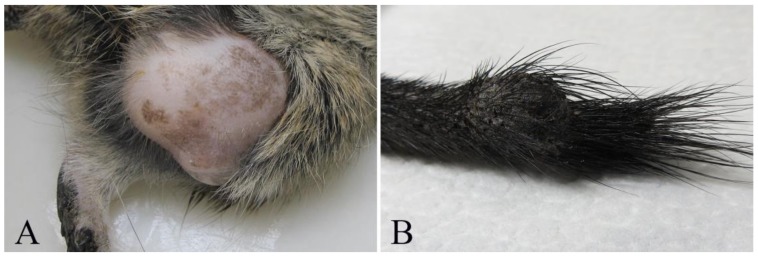
Tumors in pet degus (*Octogon degus*). (**A**) Subject # 2. Fibrosarcoma, large neoplastic mass in left hind limb. (**B**) Subject # 8. Chordoma, firm mass on the tail, measuring 1 cm in diameter.

**Figure 2 vetsci-07-00032-f002:**
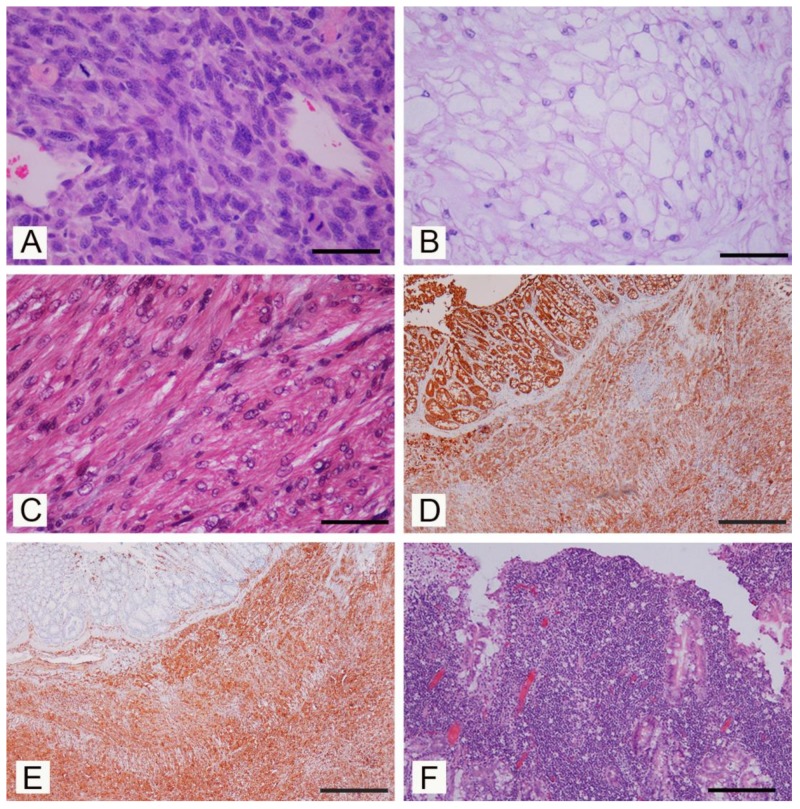
Neoplasia in pet degus (*Octogon degus*), histopathological aspects. (**A**) Fibrosarcoma, bundles of spindle cells with anisocytosis and anisokaryosis, nuclear atypia, and several mitoses (H-E; Bar = 150 µm). (**B**) Chordoma, characteristic physaliferous cells with vacuolated cytoplasm and round, eccentric nuclei (H-E; Bar = 150 µm). (**C**) Leiomyosarcoma of the urinary bladder. Bundles of neoplastic spindle smooth muscle cells with cytoplasm stained red (Trichrome Goldner stain; Bar = 150 µm). (**D**) Mesenteric mesothelioma. Neoplastic cells show marked immunoreactivity for pancytokeratin. (Immunoperoxidase stain, hematoxylin counterstain; Bar = 1000 µm). (**E**) Mesenteric mesothelioma. Neoplastic cells were also immunorecative for vimentin. (Immunoperoxidase stain, hematoxylin counterstain; Bar = 1000 µm). (**F**) Intestinal lymphoma, dense infiltrations of lymphoid cells changing the normal architecture of the intestine wall (H-E; Bar = 300 µm).

**Table 1 vetsci-07-00032-t001:** Data about gender, age, diagnosis and location of tumors and non-neoplastic lesions observed in the 16 degus examined.

Degu No.	Gender (M = Male, F = Female)	Age (Years)	Malignant Tumors/Location	Benign Tumors and Non-Neoplastic Proliferative Lesions/Location
**1**	F	6	Fibrosarcoma/Tail	
**2**	F	5	Fibrosarcoma/Hind limb	
**3**	M	5	Fibrosarcoma/Hind limb	
**4**	F	3	Fibrosarcoma/Ear	
**5**	F	8	Fibrosarcoma/Hind limb	Lipoma/Subcutis flank
**6**	F	9	Fibrosarcoma/Hind limb	
**7**	F	6	Fibrosarcoma/Back region	
**8**	M	5	Chordoma/Tail	
**9**	F	5	Osteosarcoma/Femur	
**10**	F	4	Leiomyosarcoma/Uterus and urinary bladder	
**11**	F	8	Mesothelioma/Mesentery	Hemangioma/Kidney and Nodular hyperplasia/Liver
**12**	ND	3		Squamous papilloma/Oral cavity
**13**	F	6	Lymphoma/Intestine	
**14**	M	7		Adenoma/Liver
**15**	M	8		Biliary cystoadenoma/Liver
**16**	M	8		Biliary cystoadenoma/Liver

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
