# Peer review of "Spontaneous Tumors and Non-Neoplastic Proliferative Lesions in Pet Degus (Octodon degus)"

_vetsci, 2020, doi:10.3390/vetsci7010032_

Round 1

Reviewer 1 Report

The authors describe 12 malignant tumors, 7 benign tumors, and 6 non-neoplastic proliferative changes in a total of 18 pet degus. This study constitutes a large case series of neoplasms in this animal species, in which the incidence of tumors was reported to be very low. Indeed, two previously published case series reported 7 tumors in 189 degus and 6 tumors in 300 degus.

The manuscript is well written and very well illustrated, and provides a gross and histologic description of each tumor or tumor-like lesion.

Minor revisions suggested

Throughout the manuscript: the presentation by organ system is less informative for the reader than would be a classification into malignant neoplasms, benign tumors, and non-neoplastic proliferative lesions. Some information is missing to understand the prevalence of degus tumors at your institute: do you mean that in the 2010-2015 period, you received 18 degus, and all of them carried at least a proliferative lesion? Or that during this period, 18 degus carried at least a proliferative lesion; in that case, how many tumor-free degus were examined during the same period? Page 1, line 4, affiliations: the affiliations 2 and 3 are inverted. Page 1, abstract: the abstract could be more informative. The authors could specify the number of malignant tumors, number of benign tumors, number of non-neoplastic proliferative lesions, as well as the number of cutaneous fibrosarcomas, leiomyosarcomas, and biliary cystadenomas (tumors which involved more than one animal). Pages 2-3, Tables 1 and 2: I think that the case numbers are not necessary and confusing, and the degus numbers are sufficient. I think that a single table would be easier to read; 1st column Degu number; 2nd column Gender; 3rd column Age; 4th column Malignant tumors (name + location); and 5th column Benign tumors and non-neoplastic proliferative lesions. One line per animal. Maybe add a 6th column indicating the cause of death (for the 14 necropsied animals) or the outcome after surgical excision (for the other 4 degus). Page 4, line 67, cutaneous and subcutaneous tumors: they involved 7/18 animals (not 8). Can you specify the mean age at diagnosis and age range for cutaneous fibrosarcomas (as you did for hepatic foci of cellular alteration)? Page 4, line 69, cutaneous and subcutaneous tumors: maybe specify “(N=4/7)” after “hind limbs”. Page 4, line 80, cutaneous and subcutaneous tumors: could you please specify the case numbers of the mothers with cutaneous fibrosarcoma, and the case numbers of their offsprings? Page 4, line 93, and page 6, line 137: the terms “anisocytotic” and “anisokaryotic” are uncommon, maybe replace by “with anisocytosis” and “with anisokaryosis”. Page 4, line 93, and page 5, line 99, chordoma: the neoplastic cells are more polygonal than round; would you qualify them as physaliferous (I would)? Page 5, line 98, legend for Fig 2: I cannot see atypical mitotic figures on the field 2A; would you rather say “with anisocytosis, anisokaryosis, nuclear atypia, and several mitoses”? Page 5, Figure 2: I would suggest to provide more illustrations, and to separate malignant from benign tumors/pseudotumors. Figure 2 could correspond to 4 examples of malignant tumors (fibrosarcoma, osteosarcoma, leiomyosarcoma, and lymphoma), and Figure 3 to benign tumors (chordoma, oral papilloma, biliary cystadenoma) and pseudotumors (hepatic focus of cellular alteration). Page 6, line 170, discussion: the numbers given here correspond to a prevalence (number of tumor-bearing degus among a total population of degus), not an incidence (number of cases per year per 1000 individuals at risk). Same line 178. Pages 6-7, lines 172-177, discussion: the list of reported tumors in degus is quite long; maybe it would be easier for the reader if you could separate malignant from benign tumors. Page 7, discussion: regarding fibrosarcomas, which were reported only once before in the literature, but constitutes the most common malignant tumor in your series, and affected genetically related animals: can you please discuss on the possible familial risk of fibrosarcomas in degus? At the end of this review are 3 interesting references in human oncology that investigated the familial risk of soft-tissue sarcomas. Page 7, line 196: “neoplastic” does not apply here, as the 10 gastrointestinal lesions also comprised non-neoplastic proliferative lesions. Page 7, line 211: the terms “same predisposition of low incidence of genitourinary lesions” are difficult to understand. Do you mean that both degus and chinchillas are predisposed to genitourinary lesions, even if they are rare? Or that degus and chinchillas both have a low incidence of genitourinary lesions? End of the manuscript: a short conclusion could be added. Page 8, lines 244-246, reference 14: the title of the chapter is missing. Page 8, lines 261-263, reference 20: the title of the chapter is missing.

Spelling / Typographical errors

Page 1, line 12, abstract: “sparse”. Page 2, line 48, pathological investigations: “micrometer” (instead of “micron”). Page 4, line 74: “fields” (plural). Page 4, line 88: “degus” (with s); maybe specify their case number (#8 and #9), and add “bone” in “non-neoplastic lesions”. Page 5, line 126, and page 6, line 150: Figure 2D was cited before Figure 2C. Page 6, line 131: “female” (singular). Page 6, lines 142, 151, and 156: “3-year-old”, “7-year-old”, and “8-year-old”. Page 7, line 197: “of” before “animals”, and “mainly (with i). Page 7, line 210: “common” instead of “often”, or “often diagnosed”.

References

Chan SH, Lim WK, Ishak NDB, Li ST, Goh WL, Tan GS, Lim KH, Teo M, Young CNC, Malik S, Tan MH, Teh JYH, Chin FKC, Kesavan S, Selvarajan S, Tan P, Teh BT, Soo KC, Farid M, Quek R, Ngeow J. Germline Mutations in Cancer Predisposition Genes are Frequent in Sporadic Sarcomas. Sci Rep. 2017 Sep 6;7(1):10660. doi: 10.1038/s41598-017-10333-x.

Farid M, Ngeow J. Sarcomas Associated With Genetic Cancer Predisposition Syndromes: A Review. Oncologist. 2016 Aug;21(8):1002-13. doi: 10.1634/theoncologist.2016-0079.

Benna C, Simioni A, Pasquali S, De Boni D, Rajendran S, Spiro G, Colombo C, Virgone C, DuBois SG, Gronchi A, Rossi CR, Mocellin S. Genetic susceptibility to bone and soft tissue sarcomas: a field synopsis and meta-analysis. Oncotarget. 2018 Apr 6;9(26):18607-18626. doi: 10.18632/oncotarget.24719.

Author Response

Thank you very much for your comments and suggestions which drastically improved the manuscript quality. Below please you will find a point-by-point response to reviewer’s comments 

Throughout the manuscript: the presentation by organ system is less informative for the reader than would be a classification into malignant neoplasms, benign tumors, and non-neoplastic proliferative lesions. Some information is missing to understand the prevalence of degus tumors at your institute: do you mean that in the 2010-2015 period, you received 18 degus, and all of them carried at least a proliferative lesion? Or that during this period, 18 degus carried at least a proliferative lesion; in that case, how many tumor-free degus were examined during the same period?

Data about the number of animals received in the 2010-2015 period has been added and analysed in the Discussion section.

Page 1, line 4, affiliations: the affiliations 2 and 3 are inverted.

The affiliations have been modified.

Page 1, abstract: the abstract could be more informative. The authors could specify the number of malignant tumors, number of benign tumors, number of non-neoplastic proliferative lesions, as well as the number of cutaneous fibrosarcomas, leiomyosarcomas, and biliary cystadenomas (tumors which involved more than one animal).

The abstract has been modified.

Pages 2-3, Tables 1 and 2: I think that the case numbers are not necessary and confusing, and the degus numbers are sufficient. I think that a single table would be easier to read; 1st column Degu number; 2nd column Gender; 3rd column Age; 4th column Malignant tumors (name + location); and 5th column Benign tumors and non-neoplastic proliferative lesions. One line per animal. Maybe add a 6th column indicating the cause of death (for the 14 necropsied animals) or the outcome after surgical excision (for the other 4 degus).

The tables have been chamged in a single Table 1.

Page 4, line 67, cutaneous and subcutaneous tumors: they involved 7/18 animals (not 8). Can you specify the mean age at diagnosis and age range for cutaneous fibrosarcomas (as you did for hepatic foci of cellular alteration)? Page 4, line 69, cutaneous and subcutaneous tumors: maybe specify “(N=4/7)” after “hind limbs”.

The text has been modified.

Page 4, line 80, cutaneous and subcutaneous tumors: could you please specify the case numbers of the mothers with cutaneous fibrosarcoma, and the case numbers of their offsprings?

Data has been added to the text.

Page 4, line 93, and page 6, line 137: the terms “anisocytotic” and “anisokaryotic” are uncommon, maybe replace by “with anisocytosis” and “with anisokaryosis”.

The text has been modified.

Page 4, line 93, and page 5, line 99, chordoma: the neoplastic cells are more polygonal than round; would you qualify them as physaliferous (I would)?

The text has been modified.

Page 5, line 98, legend for Fig 2: I cannot see atypical mitotic figures on the field 2A; would you rather say “with anisocytosis, anisokaryosis, nuclear atypia, and several mitoses”?

The legend has been modified as suggested.

Page 5, Figure 2: I would suggest to provide more illustrations, and to separate malignant from benign tumors/pseudotumors. Figure 2 could correspond to 4 examples of malignant tumors (fibrosarcoma, osteosarcoma, leiomyosarcoma, and lymphoma), and Figure 3 to benign tumors (chordoma, oral papilloma, biliary cystadenoma) and pseudotumors (hepatic focus of cellular alteration).

Figure 2 has been modified adding IHC aspects of mesenteric mesothelioma.

Page 6, line 170, discussion: the numbers given here correspond to a prevalence (number of tumor-bearing degus among a total population of degus), not an incidence (number of cases per year per 1000 individuals at risk). Same line 178. Pages 6-7, lines 172-177, discussion: the list of reported tumors in degus is quite long; maybe it would be easier for the reader if you could separate malignant from benign tumors.

The text has been modified.

Page 7, discussion: regarding fibrosarcomas, which were reported only once before in the literature, but constitutes the most common malignant tumor in your series, and affected genetically related animals: can you please discuss on the possible familial risk of fibrosarcomas in degus? At the end of this review are 3 interesting references in human oncology that investigated the familial risk of soft-tissue sarcomas.

Suggested references have been added.

Page 7, line 196: “neoplastic” does not apply here, as the 10 gastrointestinal lesions also comprised non-neoplastic proliferative lesions. Page 7, line 211: the terms “same predisposition of low incidence of genitourinary lesions” are difficult to understand. Do you mean that both degus and chinchillas are predisposed to genitourinary lesions, even if they are rare? Or that degus and chinchillas both have a low incidence of genitourinary lesions? End of the manuscript: a short conclusion could be added. Page 8, lines 244-246, reference 14: the title of the chapter is missing.

A short "Conclusion" section has been added to the text.

Page 8, lines 261-263, reference 20: the title of the chapter is missing.

Refernce has been modified.

Spelling / Typographical errors

The different typing errors have been modified.

Reviewer 2 Report

The manuscript entitled “Spontaneous tumors and non-neoplastic proliferative lesions in pet degus” is well-written and includes relevant general data about neoplastic lesions in pet degus.

There are, however, some points that need to be addressed:

- Reference 15 regards only epithelial and melanocytic tumors. Information concerning histological classification of urinary, reproductive and alimentary tract tumors may be found in other volumes from the same WHO collection.

- Macroscopic characterization of the lesions is poor; indeed, several parameters are missing to obtain a complete description of the tumors (dimensions, color and texture, among others)

- The order of Fig 2c and 2d must be changed to match the order in which they appear in the text.

- The reviewer has serious doubts concerning the classification of "hepatic foci of cellular alteration" as proliferative lesions. In fact, karyomegaly, anisokaryosis, cytomegaly and binucleated/multinuleated hepatocytes are common incidental findings in aging rodents. These changes may represent non-specific adaptive response to liver injury or occur spontaneously, and are not necessarily proliferative conditions.  I would suggest that the authors consider on their exclusion from this study.

- Regarding cases 11 and 12 (uterine and urinary bladder leiomyosarcoma, respectively, in the same animal), did the authors consider the possibility of being only one lesion (with infiltration of neighboring anatomical structures)?

- Regarding the lesion diagnosed as mesenteric fibrosarcoma, did the authors consider mesothelioma and gastrointestinal stromal tumor as putative differential diagnosis? In the reviewer’s opinion this possibility should be investigated using immunohistochemistry, if necessary.

- In the reviewer's opinion the manuscript lacks a conclusion.

Moreover, there are several minor details that deserve the author’s attention:

- Line 47: “from” instead of “form”

- Line 67: 7/18 (if considering individuals) or 8/25 (when considering individual lesions) subjects

- Lines 74-75: change to “…and a variable extension of necrotic areas”

- Line 85: “…as an oval…” instead of “…with an oval…”

- Line 93: “…composed of round cells with vacuolated cytoplasm exhibiting anisocytosis…” instead of “…consisted of anicocytotic…”

- Line 107: “revealed a poorly…” instead of “revealed poorly…”

- Line 111: please change to: “No osteoid production was observed…”

- Line 114: “…sarcomatous pleomorphic neoplastic cells…” instead of “polymorphism”

- Line 119: please change to: “…a well-circumscribed…mass composed of…muscle cells was observed”

- Line 121: “pleomorphism” instead of “polymorphism”

- Line 124: please change to: “Histological examination revealed an infiltrative…”

- Line 124: “…marked nuclear plemorphism…”

- Line 135: “…a large neoplastic…”

- Line 136: “..an infiltrative, non-encapsulated…”

- Lines 137-138: please change to: “Neoplastic cells, which exhibited anisocytosis and anisokaryosis, showed…”

- Line 139: “… a round …”

- Linhe 140: please change to: “Histological examination revealed a round…”

- Line 142: “An oral squamous papilloma…”

- Line 145: delete “with” please

- Line 153: “A well-circumscribed…lesion consisting of…”

- Line 171: please change to: “…in a study comprising laboaratory degus…”

- Line 172: “…300 pet degus”

- Lines 181-182 must be rewritten

- Line 186: the authors cannot stat that “females seem to be predisposed”. They can only state that there were more cases diagnosed in females than in males.

- Line 193: delete “they are”, please

- Line 205: “…resembles that described in other…”

- Line 206: “…differently from descriptions of rat chordoma…”

- Line 210-212: once again, based on these case series, the authors cannot infer about the low predisposition of degus for genitourinary tumors.

Author Response

Thank you very much for your comments and suggestions which drastically imporved the manuscript quality. Below please you will find the response point-by-pont to the dofferent comments.

Reference 15 regards only epithelial and melanocytic tumors. Information concerning histological classification of urinary, reproductive and alimentary tract tumors may be found in other volumes from the same WHO collection.

Specific references about classification criteria have been added.

Macroscopic characterization of the lesions is poor; indeed, several parameters are missing to obtain a complete description of the tumors (dimensions, color and texture, among others)

Some data about macroscopic characteristics of lesions have been added.

The order of Fig 2c and 2d must be changed to match the order in which they appear in the text.

Figure order has been properly changed.

The reviewer has serious doubts concerning the classification of "hepatic foci of cellular alteration" as proliferative lesions. In fact, karyomegaly, anisokaryosis, cytomegaly and binucleated/multinuleated hepatocytes are common incidental findings in aging rodents. These changes may represent non-specific adaptive response to liver injury or occur spontaneously, and are not necessarily proliferative conditions.  I would suggest that the authors consider on their exclusion from this study.

This part has been excluded from the mansucript.

Regarding cases 11 and 12 (uterine and urinary bladder leiomyosarcoma, respectively, in the same animal), did the authors consider the possibility of being only one lesion (with infiltration of neighboring anatomical structures)?

This hypothesis has been considered and excluded.

Regarding the lesion diagnosed as mesenteric fibrosarcoma, did the authors consider mesothelioma and gastrointestinal stromal tumor as putative differential diagnosis? In the reviewer’s opinion this possibility should be investigated using immunohistochemistry, if necessary.

Thank for your suggestion IHC revealed that this tumour was a mesenteric mesothelioma, the txt has bennproperly changed and proper figures added.

- In the reviewer's opinion the manuscript lacks a conclusion.

A "Conclusion" section has been added and the "Discussion" section improved.

Moreover, there are several minor details that deserve the author’s attention:

All the changes proposed have been included in the text.